# Extracranial Extension of a Convexity Meningioma into the Temporal Fossa: A Rare Case Report and Literature Review

**DOI:** 10.3390/diagnostics15212810

**Published:** 2025-11-06

**Authors:** Inesa Stonkutė, Dominykas Afanasjevas, Audra Janovskienė, Mindaugas Žukauskas, Darius Pranys, Albinas Gervickas

**Affiliations:** 1Faculty of Odontology, Medical Academy, Lithuanian University of Health Sciences, Eiveniu 2, LT-50161 Kaunas, Lithuania; 2Department of Maxillofacial Surgery, Medical Academy, Hospital of Lithuanian University of Health Sciences, Eiveniu 2, LT-50161 Kaunas, Lithuania; dominykas.afanasjevas@stud.lsmu.lt (D.A.); audra.janovskiene@lsmu.lt (A.J.); mindaugas.zukauskas@lsmu.lt (M.Ž.); albinas.gervickas@lsmu.lt (A.G.); 3Department of Pathology, Medical Academy, Hospital of Lithuanian University of Health Sciences, Eiveniu 2, LT-50161 Kaunas, Lithuania; darius.pranys@kaunoklinikos.lt

**Keywords:** meningioma, extracranial tumor, temporal fossa, surgical excision

## Abstract

**Background and Clinical Significance**: Meningiomas are among the most common primary intracranial tumors, usually benign and slow-growing. Extracranial extension is exceptionally rare, particularly when arising from convexity meningiomas extending into the temporal fossa. Such cases pose unique diagnostic and therapeutic challenges due to their atypical growth patterns and anatomical complexity. **Case Presentation**: A 63-year-old woman previously treated for a right temporal convexity meningioma with subtotal resection and Gamma Knife radiosurgery demonstrated progressive extracranial tumor growth over five years, while the intracranial component remained stable. MRI revealed infiltration of the temporalis and lateral pterygoid muscles and erosion of the temporal bone. Due to extensive extracranial involvement and limited neurosurgical accessibility, resection was performed by a maxillofacial surgical team through a preauricular approach. Intraoperatively, the tumor was encapsulated but adherent to the deep temporal fascia and zygomatic arch. The temporal branch of the facial nerve was identified and preserved. Histopathology confirmed a meningothelial meningioma, WHO Grade I, with low proliferative activity (Ki-67 < 1%). Postoperative recovery was uneventful, with transient facial nerve weakness that resolved within weeks. **Conclusions**: This report adds to the limited literature describing temporal fossa involvement by convexity meningiomas and illustrates the value of collaboration between neurosurgical and maxillofacial teams. Regular MRI surveillance every 6–12 months is advised for early detection of recurrence.

## 1. Introduction

Meningioma is a typically slow-growing, benign neoplasm of the central nervous system (CNS), most commonly arising from arachnoid cap cells located on the inner surface of the dura mater [1]. The arachnoid cells originate from meningeal precursor cells, which are derived from mesodermal and neural crest cell lineages [2]. Meningiomas can also manifest as extracranial masses, which are classified into primary tumors, confined entirely to the extracranial space without an associated intracranial mass, or secondary tumors, arising from the extension of an intracranial meningioma [3]. Extracranial spread can occur through natural anatomical pathways, such as foramina and skull fissures, or via direct invasion of bone and soft tissue, often accompanied by hyperostosis [4]. Additionally, prior trauma or surgical interventions, particularly when resection is incomplete or when tumor spillage occurs, may create iatrogenic pathways for extracranial tumor growth [5]. Convexity meningiomas may extend into the temporal fossa, particularly through invasion of the squamous portion of the temporal bone. As these tumors grow outward, they may involve the temporalis muscle, subgaleal soft tissue, and, in rare cases, the overlying skin [6]. Extracranial meningiomas, particularly those involving extension into the temporal fossa, are exceptionally rare, with only a limited number of documented case reports in the literature. The majority of temporal fossa meningiomas arise as extensions of primary intracranial tumors. Common routes of extension include the tegmen tympani, posterior petrous ridge, jugular bulb, internal auditory canal, and the sulcal pathways of the greater and lesser superficial petrosal nerves [7]. This report aims to present a rare case of convexity meningioma with bone involvement and extension into the temporal fossa, emphasizing clinical symptoms, differential diagnosis, treatment strategies, and surgical management.

## 2. Literature Review

### 2.1. Definition

Meningiomas are slow-developing neoplasms primarily composed of arachnoid cap cells attached to the pia mater and inner arachnoid, constituting the most common category of primary central nervous system (CNS) tumors [8]. According to the World Health Organization (WHO) classification, meningiomas are divided into grade I (benign), grade II (atypical), and grade III (malignant), with the majority being grade I benign lesions [9]. When completely resected, meningiomas typically exhibit a favorable prognosis, particularly in younger patients. They rarely extend beyond the cranial boundaries but may cause osteolytic changes in the skull or invade adjacent structures, such as the temporal fossa [2].

### 2.2. Epidemiology

Meningiomas are the most common primary CNS tumors in adults, representing about 36% of all CNS neoplasms and 53% of non-malignant intracranial tumors, with an annual incidence of approximately 7.9 per 100,000 individuals [2]. Extracranial extension occurs in only 6–20% of cases, most often through natural skull foramina or bone invasion. However, involvement of the temporal or infratemporal fossa is exceedingly uncommon—accounting for roughly 2% of extracranial meningiomas and <0.5% of all meningiomas [1,7,10]. In their recent systematic review of temporal bone and fossa meningiomas, Burato et al. (2024) identified fewer than 40 reported cases worldwide, with a clear female predominance and mean age around 55 years [10].

### 2.3. Symptoms

Meningiomas may remain asymptomatic for extended periods, often existing in the brain without causing noticeable effects for several years. When symptoms do occur, they develop gradually due to compression of adjacent structures rather than direct invasion [2]. Clinical presentation of meningiomas are largely determined by the size and anatomical location of the lesion. Neurological symptoms may include seizures, focal motor activity such as isolated or multifocal muscle twitches or spasms, and alterations in autonomic control. Additionally, patients may experience sensory disturbances, partial or complete loss of consciousness, and other deficits corresponding to the affected cerebral regions. Extracranial meningiomas, particularly those extending into the temporal fossa, are rare and frequently misdiagnosed due to non-specific signs [11,12].

Extracranial meningiomas can cause bone erosion or perforation, particularly when they infiltrate the temporal bone, tegmen tympani, or mastoid structures [13]. In cases where both the bone and dura mater are breached, a cerebrospinal fluid (CSF) leak may occur, potentially presenting as clear otorrhea and carrying the risk of complications such as meningitis [14].

Patients may present with auditory and vestibular symptoms such as hearing loss, otalgia, tinnitus, a sensation of ear fullness and dizziness [15]. In rarer cases, a temporal fossa meningioma can significantly affect the trigeminal nerve, particularly its mandibular division (V3). Mandibular nerve exits the skull via the foramen ovale into the infratemporal fossa, placing it in direct proximity to tumors in this region [16]. This involvement may result in facial hypoesthesia, paresthesia of the lower face, jaw, and anterior tongue, as well as weakness of the muscles of mastication. Additionally, patients may present with trigeminal neuralgia-like pain, which is characterized by sharp, shooting pain in the mandibular region [17,18].

Due to their proximity to the temporomandibular joint (TMJ), temporal fossa meningiomas can mimic TMJ disorders. Overlapping symptoms include pain in the jaw or preauricular area, trismus, difficulty chewing, and jaw deviation on opening. These findings are typically the result of tumor-induced mass effect on adjacent muscles and neural pathways, potentially delaying accurate diagnosis [19].

Moreover, temporal fossa meningioma can affect the eyeball through mass effect or extension into the orbit. This may lead to proptosis when the tumor compresses or invades the orbital cavity, especially if it extends anteromedially through structures like the lateral orbital wall [20]. Involvement of the extraocular muscles or cranial nerves III, IV, and VI may impair coordinated eye movements, causing diplopia. Medial extension toward the optic canal can compress the optic nerve (CN II), potentially leading to visual field defects or optic atrophy, resulting in progressive vision loss. Patients may also experience orbital pain or a sensation of pressure from tumor compression of surrounding tissues [3]. In some cases, tumor extension into the lacrimal gland can disrupt tear production, causing ocular dryness and discomfort [21].

### 2.4. Differentiation

The differential diagnosis of meningioma entails distinguishing it from a variety of intracranial and extracranial pathologies that may exhibit overlapping clinical, radiological, or histopathological characteristics. In particular, the differential diagnosis of secondary temporal fossa meningiomas includes schwannomas, cholesteatomas, glomus tumors, metastatic lesions, gliomas, lymphomas, and infectious conditions such as osteomyelitis and chronic otitis media [22].

Radiologically, the differentiation is based on distinctive features such as enhancement patterns, the shape and location of the mass, and the presence or absence of dural attachment. Meningiomas typically demonstrate a well-circumscribed mass with homogeneous contrast enhancement and a characteristic “dural tail” sign, whereas other conditions may exhibit more irregular, infiltrative growth, heterogeneous enhancement, or significant vascular involvement [23].

Benign temporal fossa meningiomas are characterized by distinct histological features that aid in their identification and differentiation. A key diagnostic trait of meningiomas is the whorled growth pattern, in which tumor cells form concentric, spiral-like arrangements. Additionally, the presence of psammoma bodies, calcified, concentric, laminated structures, further distinguishes meningiomas from gliomas or schwannomas, which do not contain these calcifications [24]. The tumor cells are typically polygonal or epithelioid, with round nuclei and eosinophilic cytoplasm. Benign meningiomas also exhibit low mitotic activity and minimal necrosis, features that help differentiate them from more aggressive tumors like glioblastomas or atypical and malignant meningiomas, which display higher mitotic rates and more pronounced necrosis [25]. Vascularity is often prominent, though not exclusive to temporal fossa meningiomas, as it can also be seen in other highly vascular tumors such as hemangiopericytomas [26].

### 2.5. Diagnostic Criteria

Meningiomas are typically diagnosed based on clinical presentation, imaging, and histopathology. MRI is the gold standard for diagnosis, owing to its superior soft tissue contrast and multiplanar capabilities [27]. Temporal fossa meningiomas typically present as well-circumscribed, extra-axial masses that are isointense to gray matter on T1-weighted sequences and isointense to mildly hyperintense on T2-weighted images. Post-contrast T1-weighted imaging characteristically reveals intense, homogeneous enhancement, often accompanied by the “dural tail” sign—a finding indicative of dural involvement or reactive changes. MRI also provides critical information about tumor extension into adjacent compartments.

CT is particularly effective for evaluating osseous involvement. Hallmark features include hyperostosis and bone erosion at the site of dural attachment, most commonly affecting the temporal bone and skull base. CT is also highly sensitive in detecting intratumoral calcifications such as psammoma bodies, which appear as hyperdense foci on non-contrast scans [27,28].

Histological examination confirms the diagnosis and informs prognosis and treatment planning. Histopathologically, temporal fossa meningiomas resemble their intracranial counterparts. They are typically composed of meningothelial cells arranged in sheets or whorls, often with the presence of psammoma bodies [29].

Immunohistochemistry plays a key role in diagnosing Grade I meningiomas. Important markers include vimentin, epithelial membrane antigen (EMA), progesterone receptor (PR), and S-100. A low Ki-67 (MIB-1) index, typically less than 2–3%, indicates low proliferative activity, which is consistent with the benign nature of Grade I meningiomas [30,31].

### 2.6. Treatment

Meningioma treatment generally involves surgical resection for symptomatic or enlarging tumors, with the goal of achieving maximal safe resection while preserving neurological function [32]. The extent of resection (EOR) plays a critical role in prognosis, with classifications of gross total resection (GTR) or subtotal resection (STR), depending on the tumor’s location and proximity to vital structures. GTR is preferred when feasible, as it provides the best chance of long-term control, whereas STR is considered when complete removal would pose significant risk to neurological function [33].

Postoperative radiation therapy, including options like stereotactic radiosurgery (SRS) or conformal fractionated radiotherapy (RT), is used for residual or recurrent tumors [27,34,35]. The treatment of extracranial meningiomas, particularly WHO Grade I lesions, generally follows similar principles. Gross total surgical resection (GTR) is the standard of care and is often curative, especially since these tumors grow slowly and have a low recurrence potential [35]. Extracranial meningiomas typically carry less risk of brain injury compared to their intracranial counterparts, as their location outside the cranial cavity allows for safer, more complete resections. This advantage facilitates enhanced oncologic effectiveness and reduces the risk to functional brain tissue [36].

Adjuvant radiotherapy is considered when complete excision is not achievable, when the tumor recurs, or when pathological findings indicate more aggressive features [10]. However, for fully resected WHO Grade I tumors, radiation therapy is not routinely indicated [36]. At present, there is no established role for systemic therapies, such as chemotherapy or targeted agents, in the management of Grade I meningiomas, as these tumors tend to be resistant to pharmacologic treatment [27,37]. Postoperative surveillance with MRI is crucial, even for fully resected lesions, typically starting every 6 to 12 months and later transitioning to annual follow-ups [36]. Early detection of recurrence is key to ensuring favorable outcomes. Overall, the prognosis following complete resection of an extracranial WHO Grade I meningioma, especially in locations such as the temporal fossa, is excellent, with very low long-term recurrence rates [38].

## 3. Case Presentation

A 63-year-old female was referred to the Department of Oral and Maxillofacial Surgery at the Hospital of the Lithuanian University of Health Sciences Kaunas Clinics with a complaint of a painless, mobile mass in the right temporal fossa. Her medical history was notable for a right temporal convexity meningioma, initially treated in 2019 at Aberdeen Hospital, Scotland. Prior to surgical intervention in 2019, she experienced progressive neurological deterioration characterized by dizziness, unsteady gait, vertigo, occipital headaches, nausea, upper limb weakness, distal paresthesia, behavioral disturbances including agitation and insomnia, as well as right ear pain.

Magnetic resonance imaging performed in November 2019 revealed a right temporal meningioma exhibiting both intracranial and extracranial extension, with probable involvement of the right mastoid region. Following preoperative planning and custom cranial plate fabrication, she underwent craniotomy and subtotal resection of the right temporal and extracranial tumour in December 2019. Intraoperatively, a large skull base-infiltrating meningioma was confirmed. Histopathology revealed a WHO Grade I meningioma. Resection was Simpson Grade III due to residual tumour within the skull base that was deemed unresectable.

The postoperative course was complicated by transient neuropsychiatric symptoms, including visual and auditory hallucinations, persecutory delusions, agitation, insomnia, and cognitive impairment. These were likely multifactorial in origin, attributed to the postoperative state, corticosteroid therapy, and levetiracetam initiated preoperatively. Notably, these neuropsychiatric symptoms resolved spontaneously without targeted pharmacological intervention. By January 2020, the patient had achieved neurological stability and was asymptomatic. However, ongoing surveillance with annual MRI demonstrated slow but progressive tumour recurrence, predominantly localized to the temporal fossa.

In October 2021, the patient was hospitalized for a neurosurgical evaluation following further tumor growth. Neurological examination at that time revealed no deficits: the patient was conscious, oriented, with intact cranial nerve function, full limb strength, and preserved coordination and reflexes. A multidisciplinary oncology team reviewed the case and recommended stereotactic radiosurgery. The patient underwent Gamma Knife treatment with 14 Gy at the 50% isodose.

She was discharged post-procedure with recommendations for analgesics, neurosurgical follow-up in six months, and MRI in one year. Imaging performed in September 2022 showed stable findings without intracranial progression. However, a follow-up MRI in September 2023 indicated slight enlargement of the extracranial tumor component, by approximately 3–4 mm in all directions, while the intracranial component remained unchanged. The patient remained asymptomatic. In November 2023, a second Gamma Knife procedure was performed, delivering 13 Gy at the 53% isodose. The extracranial component of the mass was not addressed during this session.

By September 2024, MRI demonstrated further progression of the extracranial mass, now measuring approximately 36 mm anteroposteriorly, 18 mm transversely, and 49 mm craniocaudally. Imaging showed thickened dura (4–6 mm) with likely residual tumor at the anterior base of the right temporal lobe, with associated bone infiltration involving the petrous part of the temporal bone. Postoperative changes, including cystic gliosis and hemosiderin deposits, were evident, and the right mastoid air cells remained fluid filled. Additionally, a left-sided calcified extra-axial meningioma (approximately 12 mm in diameter) was observed. The patient again reported no symptoms. Neurosurgical consultation in December 2024 recommended surgical excision of the extracranial portion of the tumor. During a follow-up visit in February 2025, the neurosurgical team concluded that further resection was not feasible due to the extracranial tumor component extending beyond the field accessible via craniotomy. The lesion involved the temporalis muscle, mandibular notch, and lateral pterygoid space—areas more safely approached through extracranial routes familiar to maxillofacial surgeons. Consequently, a maxillofacial approach was selected to minimize risk to intracranial structures and the middle cranial fossa while allowing optimal visualization and preservation of the facial nerve.

On 18 February 2025, the patient presented to the Department of Oral and Maxillofacial Surgery at the Hospital of the Lithuanian University of Health Sciences Kaunas Clinics for a consultation with a maxillofacial surgeon regarding a lesion in the temporal region, specifically in the temporal fossa. On examination, a mobile, painless node was observed in the temporal fossa. Facial nerve function was intact, and no pathological changes were observed intraorally. The patient had a history of thyroid hypertrophy and was taking medication for thyroiditis. She also had arterial hypertension, treated with Perindopril arginine 5 mg and a combination of Bisoprolol fumarate/Perindopril arginine 5 mg/5 mg. She denied any medication allergies. For further evaluation, an MRI scan and core needle biopsy of the lesion were ordered during the consultation.

On 18 March 2025, the patient returned for a follow-up consultation with the maxillofacial surgeon following completion of the MRI and core biopsy.

A core needle biopsy was performed using ultrasound guidance and local anesthesia. Three samples were taken using an 18G needle; however, histological examination showed that the obtained material was of low informational value—composed of fibrotic striated muscle tissue with a minimal amount of inflammatory infiltrates.

MRI performed in March 2025 showed persistent residual tumor extending from the anterior base of the middle cranial fossa into the petrous temporal bone, infiltrating the tympanic cavity, mastoid process, and encasing ossicles. The mass extended into the external auditory canal, measuring 2.2 cm × 1.4 cm × 1.0 cm. The extracranial tumor had slightly enlarged to 5.2 cm × 2.8 cm × 3.2 cm and demonstrated invasion of the temporalis muscle, mandibular notch, lateral pterygoid muscle, and greater wing of the sphenoid. There were also signs of encephalomalacia in the right temporal lobe, persistent mastoid opacification, and radiologic features consistent with “silent sinus syndrome”, a condition characterized by progressive maxillary sinus atelectasis and orbital floor retraction secondary to chronic negative sinus pressure (Figure 1) [15].

Based on the clinical presentation and radiological findings, a diagnosis of extracranial meningioma in the right temporal region was established. Consequently, the patient was scheduled for a surgical excision of the extracranial tumor at the Department of Oral and Maxillofacial Surgery.

Surgical excision was performed on 28 March 2025, under general anesthesia. The surgical approach involved a preauricular incision extending superiorly along the hairline to provide access to the temporal fossa while minimizing aesthetic compromise. After careful dissection through the subcutaneous tissue and superficial temporal fascia, the temporal branch of the facial nerve was identified and preserved using intraoperative nerve monitoring. The temporalis muscle was partially detached and retracted to expose the tumor, which was found to be well-encapsulated but adherent to the zygomatic arch and lateral pterygoid fascia. The lesion was meticulously dissected from surrounding musculature and bone using blunt and sharp dissection, achieving complete macroscopic excision. Intraoperative findings included moderate vascularity and firm adherence to the deep temporal fascia, without intracranial communication. Hemostasis was secured using bipolar cautery, and layered closure was performed. The excised specimen was submitted for histopathological examination.

The patient was discharged on 31 March 2025, in good general condition. She reported no complaints apart from mild right-sided swelling and small hematomas. Persistent dysfunction of the temporal branch of the facial nerve was observed. Postoperatively, she received prophylactic antibiotics and nonsteroidal anti-inflammatory drugs (NSAIDs) for pain management. A referral to a rehabilitation specialist was made to support facial nerve recovery.

Histological evaluation confirmed a diagnosis of meningothelial meningioma, WHO Grade I (M9531/0). Gross examination revealed a 7 cm × 3.2 cm × 3.0 cm soft tissue specimen containing a well-defined, firm white nodule (3.3 cm × 2.5 cm × 1.7 cm) with a grayish, elastic cut surface. Microscopically, the lesion consisted of monomorphic meningothelial cells arranged in whorls and sheets, embedded in fibrous and skeletal muscle tissue. Immunohistochemistry demonstrated EMA positivity, and the Ki-67 proliferation index was low (<1%) (Figure 2).

The patient was advised to undergo periodic clinical and radiological evaluations for systematic follow-up, given the atypical presentation of the meningioma, which warranted documentation of this case.

Postoperatively, the patient was enrolled in a structured radiological surveillance program, with MRI scheduled every six months for the first two years and annually thereafter if no progression is detected. At around six months, the patient was in good general condition, demonstrating complete restoration of facial symmetry and function. No neurological or masticatory abnormalities were identified, and she remained free of pain or sensory disturbances. A prior course of rehabilitation had successfully addressed right temporal branch facial nerve dysfunction. Ongoing neurosurgical follow-up is planned to monitor for potential recurrence.

## 4. Discussion

Extracranial extension of meningiomas is an uncommon occurrence, particularly in convexity meningiomas extending into the temporal fossa [7]. This case illustrates a rare presentation of a recurrent WHO Grade I meningioma demonstrating progressive extracranial invasion, ultimately necessitating surgical intervention by an oral and maxillofacial surgical team—a management approach infrequently documented in the literature.

Unlike previously reported temporal bone or spheno-orbital meningiomas, this lesion originated from a convexity site and demonstrated gradual extracranial progression over five years following subtotal resection and radiosurgery. The pathway of extension—through bone erosion of the temporal squama into the masticator space—differs from the more typical skull-base foraminal spread described in the literature [7,9,10,21]. This emphasizes that even histologically benign convexity meningiomas can follow an atypical outward growth pattern and require multidisciplinary planning that includes maxillofacial expertise for complete extracranial control.

The patient presented with a mobile, painless mass in the temporal region and remained asymptomatic despite significant extracranial involvement. This is consistent with the typically indolent clinical course of WHO Grade I meningiomas. Notably, there were no trigeminal, facial, auditory, or vestibular deficits despite radiological evidence of tumor infiltration into the mandibular notch, temporalis and lateral pterygoid muscles, and tympanic cavity with ossicular encasement. In contrast to other reports describing symptoms such as jaw pain, trismus, or hearing disturbances, this case underscores the diagnostic challenge posed by extracranial meningiomas, which may remain clinically silent until reaching considerable size [15,17,18].

Radiologically, the lesion demonstrated hallmark features of meningiomas, including homogeneous contrast enhancement, dural thickening, and bone hyperostosis. Although these findings are typically associated with benign histology, the extensive osseous involvement of the petrous temporal bone and greater wing of the sphenoid suggested locally aggressive behavior [27,28,29]. Histopathological evaluation confirmed a meningothelial WHO Grade I meningioma with a low Ki-67 index (<1%), reinforcing that benign meningiomas can still exhibit invasive potential without malignant transformation [30,39].

Radiologic differentials for temporal fossa masses include skull-base osteomas, paragangliomas, schwannomas, and metastatic deposits. Meningiomas typically show homogeneous enhancement, a dural tail, and associated hyperostosis on MRI and CT, features that help distinguish them from other lesions [27,28,29,30]. Osteomas appear as uniformly hyperdense, well-circumscribed bony outgrowths without a soft-tissue component or contrast enhancement [27,29], whereas metastatic lesions often demonstrate irregular margins, heterogeneous enhancement, and adjacent bone destruction [29,30]. Schwannomas may show cystic changes, heterogeneous enhancement, and expansion of neural foramina, while paragangliomas display intense enhancement with characteristic “salt-and-pepper” vascular signal voids [16,17,18,23]. Recognizing these distinguishing radiologic patterns aids differentiation of atypical extracranial meningiomas from other temporal fossa masses (Table 1).

Several previous reports have described extracranial extension of WHO Grade I meningiomas, particularly en plaque or spheno-orbital variants, demonstrating aggressive local invasion despite benign histology [9,21]. However, extracranial progression of a convexity meningioma into the temporal fossa remains exceedingly rare. In the multicenter review by Han et al. (2021), temporal bone involvement was observed in only a small subset of cases, and none exhibited extensive extracranial spread following stereotactic radiosurgery [7]. Similarly, Burato et al. (2024), in their systematic review of temporal bone meningiomas, identified fewer than 40 documented cases worldwide, none arising from convexity lesions [10]. This case therefore represents an unusual instance of locally aggressive yet histologically benign behavior, emphasizing that the proliferation index alone may not fully predict biological potential.

The standard treatment for WHO Grade I meningiomas involves surgical resection, with gross total resection (GTR) offering the best long-term outcomes. However, in cases involving the skull base or critical neurovascular structures, subtotal resection (STR) may be necessary, often followed by stereotactic radiosurgery (SRS) to manage residual disease [33,34]. Although extracranial meningiomas are rare, they are typically managed using the same principles, with resection often more feasible due to their extracranial location [35].

In this case, the patient underwent STR of a right temporal meningioma with both intracranial and extracranial components in 2019. Because of skull base involvement, GTR was not feasible. Tumor progression was later observed predominantly in the extracranial compartment. Two courses of Gamma Knife SRS (14 Gy in 2021 and 13 Gy in 2023) targeted the intracranial component only.

As the extracranial tumor progressed, the patient was referred to an oral and maxillofacial surgeon. Complete resection of the extracranial mass was achieved in 2025 via a preauricular approach. Postoperative recovery was uneventful except for transient dysfunction of the temporal branch of the facial nerve—a recognized complication of preauricular and temporal surgical approaches. Given the close anatomical relationship between the extracranial tumor and the facial nerve’s peripheral branches, particularly the temporal branch, transient neuropraxia was anticipated and resolved during follow-up [40]. Histological analysis again confirmed a meningothelial WHO Grade I meningioma.

Postoperatively, the patient was enrolled in a structured radiological surveillance program, with MRI every six months for the first two years and annually thereafter if no progression is detected, in accordance with the European Association of Neuro-Oncology (EANO) guidelines [27]. The multidisciplinary follow-up team includes neurosurgery, maxillofacial surgery, and radiology specialists, ensuring coordinated evaluation of both intracranial and extracranial compartments. Re-intervention will be considered if radiological progression exceeds 20% in any dimension or if new neurological or maxillofacial symptoms develop.

## 5. Conclusions

This case report underscores the importance of long-term clinical and radiological surveillance in patients with residual or recurrent meningioma, particularly those with prior subtotal resection and radiosurgical intervention. The gradual extracranial progression of a WHO Grade I convexity meningioma into the temporal fossa, with involvement of adjacent soft tissues and osseous structures, highlights the potential for atypical growth patterns. Accurate diagnosis relies on a multidisciplinary approach integrating detailed imaging, histopathological analysis, and clinical assessment to guide timely and appropriate management.

## Figures and Tables

**Figure 1 diagnostics-15-02810-f001:**
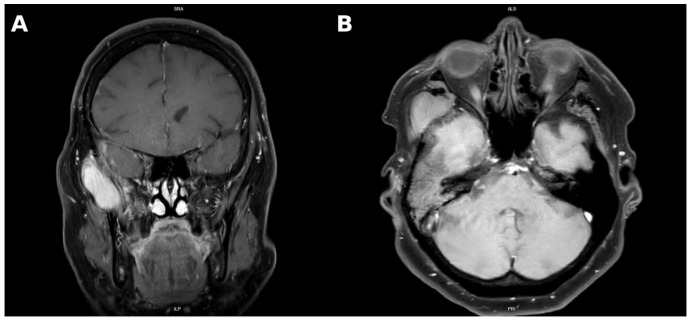
Coronal contrast-enhanced T1-weighted MRI demonstrating a homogeneously enhancing mass localized in the temporal and infratemporal fossa (**A**). Axial contrast-enhanced T1-weighted MRI demonstrating a homogeneously enhancing soft tissue mass in the right infratemporal and masticator space, with infiltration of the lateral pterygoid and temporalis muscles and compression of the right external auditory canal (**B**).

**Figure 2 diagnostics-15-02810-f002:**
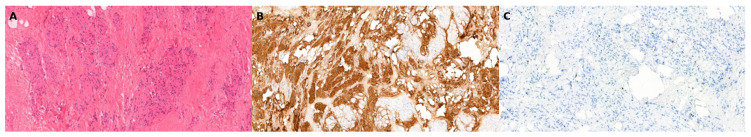
Microscopic examination of the resected soft tissue lesion stained with hematoxylin and eosin (H&E) shows monomorphic meningothelial cells arranged in whorls and sheets, embedded within fibrous tissue and skeletal muscle (**A**, 10×). Immunohistochemical staining demonstrates tumor cell positivity for epithelial membrane antigen (EMA) (**B**, 4×) and a low Ki-67 proliferation index, indicating low mitotic activity (**C**, 10×).

**Table 1 diagnostics-15-02810-t001:** Radiologic differentiation of temporal fossa lesions.

Lesion	Typical Imaging Features	Key Differentiator
**Meningioma**	Iso- to hypointense on T1; strong homogeneous enhancement; dural tail; hyperostosis	Extra-axial origin, uniform enhancement
**Osteoma**	Dense, well-circumscribed bony mass on CT; no soft-tissue component	Purely osseous lesion
**Schwannoma**	Heterogeneous enhancement; cystic areas; nerve canal widening	No dural attachment
**Metastasis**	Irregular margins; bone destruction; heterogeneous signal	Discontinuous with dura
**Glomus tumor**	Intense “salt-and-pepper” enhancement	Flow voids, vascular nature

## Data Availability

The data presented in this study are available on reasonable request from the corresponding author. The data are not publicly available due to patient privacy and ethical restrictions. Representative anonymized imaging and histological figures are included within the article.

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
