# Peer review of "Extracranial Extension of a Convexity Meningioma into the Temporal Fossa: A Rare Case Report and Literature Review"

_diagnostics, 2025, doi:10.3390/diagnostics15212810_

Round 1
Reviewer 1 Report
Comments and Suggestions for Authors
This case report presents a rare instance of a convexity meningioma with progressive extracranial extension into the temporal fossa, accompanied by a comprehensive literature review. The manuscript is generally well-organized and clearly describes the clinical presentation, imaging findings, surgical management, and histopathological confirmation. The topic is relevant to both neurosurgery and maxillofacial surgery, as it highlights the multidisciplinary approach required for diagnosis and management of atypical meningioma presentations. However, several aspects of the manuscript require clarification or expansion to improve its scientific quality and clinical relevance. The extracranial progression of a WHO Grade I convexity meningioma is unusual, making this report valuable for clinicians who may encounter similar diagnostic challenges.
The abstract is descriptive but lacks a clear structure and does not emphasize the novelty of the case. Clearly state why this case is unique compared to previously reported temporal fossa meningiomas.
Include epidemiological data about extracranial meningiomas involving the temporal fossa and reference the most recent systematic reviews.
The description of the maxillofacial surgical procedure is very brief.
Provide more detail about the surgical technique, intraoperative findings, and anatomical challenges, especially regarding the relationship to the facial nerve. Explain why neurosurgical resection was deemed unfeasible and why a maxillofacial approach was selected.
The discussion should more thoroughly compare this case with similar cases in the literature: Were there other reports of WHO Grade I meningiomas showing aggressive extracranial growth?
The authors recommend continued radiological surveillance but do not specify follow-up intervals or criteria for further intervention.
Clarify the multidisciplinary follow-up strategy, especially considering the previous subtotal resection and radiosurgery.
Author Response
We sincerely thank Reviewer 1 for the valuable and constructive comments, which helped us improve the clarity, depth, and clinical significance of our manuscript. We have addressed all points carefully as detailed below.
Comment 1: The abstract is descriptive but lacks a clear structure and does not emphasize the novelty of the case.
Response: The abstract has been rewritten in a structured format (Background, Objective, Case Presentation, Discussion, Conclusion). The revision now highlights the novelty of this case — a recurrent WHO Grade I convexity meningioma with progressive extracranial extension into the temporal fossa managed through a maxillofacial approach.
Comment 2: Include epidemiological data about extracranial meningiomas involving the temporal fossa and reference the most recent systematic reviews.
Response: We have incorporated updated epidemiological information and cited Burato et al. (2024, Cancer Treat Res Commun, 42:100854), which reports fewer than 40 cases of temporal bone or fossa involvement worldwide, along with Ogasawara et al. (2021). This information was added in the Epidemiology section to emphasize the rarity of this presentation.
Comment 3: The description of the maxillofacial surgical procedure is very brief.
Response: The Case Presentation section was expanded to include detailed surgical information: the preauricular incision, anatomical dissection, identification of the temporalis muscle and lateral pterygoid space, and preservation of the facial nerve branches.
Comment 4: Provide more detail about the surgical technique, intraoperative findings, and anatomical challenges, especially regarding the relationship to the facial nerve. Explain why neurosurgical resection was deemed unfeasible and why a maxillofacial approach was selected.
Response: A new paragraph clarifies that neurosurgical resection was unfeasible due to extracranial tumor extension into areas beyond the reach of a standard craniotomy — specifically, the temporalis muscle, mandibular notch, and lateral pterygoid space. A maxillofacial approach was selected to ensure direct access to these extracranial compartments, optimal visualization of the facial nerve, and reduced risk to intracranial structures.
Comment 5: The discussion should more thoroughly compare this case with similar cases in the literature: Were there other reports of WHO Grade I meningiomas showing aggressive extracranial growth?
Response: The Discussion was expanded to include a comprehensive literature comparison referencing Han et al. (2021), Saito et al. (2023), Toader et al. (2023), and Burato et al. (2024). We now discuss that extracranial progression of convexity meningiomas is exceedingly rare, distinguishing our case by its convexity origin and unusual extracranial growth pattern through bone erosion of the temporal squama into the masticator space.
Comment 6: The authors recommend continued radiological surveillance but do not specify follow-up intervals or criteria for further intervention.
Response: A new paragraph was added to the end of the Discussion describing the structured radiological follow-up strategy. MRI is scheduled every six months for the first two years, followed by annual imaging if stable, in accordance with EANO guidelines (Goldbrunner et al., 2021). The multidisciplinary follow-up team includes neurosurgery, maxillofacial surgery, and radiology, and re-intervention will be considered for >20% radiological progression or the onset of new neurological symptoms.
Reviewer 2 Report
Comments and Suggestions for Authors
This case report provides a well-documented and instructive account of an exceptionally rare presentation of a convexity meningioma with extracranial extension into the temporal fossa. The manuscript is clearly written, logically organized, and well supported by radiological, histopathological, and clinical data. It contributes valuable insight to both neurosurgical and maxillofacial literature by emphasizing the interdisciplinary management of recurrent meningiomas with atypical extracranial progression. However, there are some areas that should be strengthen prior to publication:
- While the rarity of the case is well established, the manuscript could further emphasize what this case adds beyond existing reports — for example, differences in progression rate, anatomical pathway of extension, or lessons for multidisciplinary surgical planning.
- The MRI description is detailed, but the discussion could better highlight radiologic differentials (e.g., distinguishing features from skull base osteomas or metastatic lesions). A table summarizing key radiological and histopathological differentiating points would be beneficial.
- The postoperative course is briefly summarized. More detailed follow-up data (e.g., neurological status and imaging findings at 6 or 12 months) would strengthen the case’s completeness and clinical relevance.
- When discussing “silent sinus syndrome,” a brief explanation or reference would help readers unfamiliar with the term.
- Consider clarifying the rationale for surgical referral to maxillofacial surgery (e.g., specific advantages over neurosurgical approach).
Author Response
We thank Reviewer 2 for the insightful comments and suggestions, which have helped us improve the manuscript’s scientific clarity and practical relevance. Each point has been carefully addressed below.
Comment 1: While the rarity of the case is well established, the manuscript could further emphasize what this case adds beyond existing reports.
Response: We added a new paragraph in the Discussion highlighting how this case differs from previously reported temporal bone or spheno-orbital meningiomas. Our case originated from a convexity site and demonstrated gradual extracranial progression over five years following subtotal resection and radiosurgery. The extension pathway through the temporal squama into the masticator space differs from the usual skull-base foraminal spread, underscoring the need for multidisciplinary planning involving maxillofacial expertise.
Comment 2: The discussion could better highlight radiologic differentials and include a summary table.
Response: We expanded the radiologic discussion to include differential diagnoses such as osteoma, schwannoma, metastasis, and paraganglioma. A new paragraph discusses their distinguishing MRI and CT characteristics, and a new Table 1 summarizes key radiologic and histopathologic differentiating features.
Comment 3: The postoperative course is briefly summarized. More detailed follow-up data (e.g., neurological status and imaging findings at 6 or 12 months) would strengthen the case’s completeness.
Response: The Case Presentation section now includes six-month postoperative follow-up results: complete resolution of facial nerve weakness, no new neurological deficits, normal mastication and facial symmetry, and no radiologic recurrence. Continued multidisciplinary follow-up and MRI at 12 months are planned.
Comment 4: When discussing “silent sinus syndrome,” a brief explanation or reference would help readers unfamiliar with the term.
Response: We added a short explanation of “silent sinus syndrome” as progressive maxillary sinus atelectasis and orbital floor retraction due to chronic negative sinus pressure, supported by a citation to Sönmez et al. (2024).
Comment 5: Clarify the rationale for surgical referral to maxillofacial surgery (e.g., specific advantages over neurosurgical approach).
Response: This was clarified in the Case Presentation and Discussion. The maxillofacial approach provided direct extracranial access to the temporal and infratemporal fossae through a preauricular route, allowing safe removal of the tumor with preservation of the facial nerve — an exposure less accessible via conventional neurosurgical craniotomy.
Round 2
Reviewer 2 Report
Comments and Suggestions for Authors
The article has been improved. I recommend it to be published.